# Hierarchical Clustering and Trajectory Analyses Reveal Viremia-Independent B-Cell Perturbations in HIV-2 Infection

**DOI:** 10.3390/cells11193142

**Published:** 2022-10-06

**Authors:** Emil Johansson, Priscilla F. Kerkman, Lydia Scharf, Jacob Lindman, Zsófia I. Szojka, Fredrik Månsson, Antonio Biague, Patrik Medstrand, Hans Norrgren, Marcus Buggert, Annika C. Karlsson, Mattias N. E. Forsell, Joakim Esbjörnsson, Marianne Jansson

**Affiliations:** 1Department of Laboratory Medicine, Lund University, 22184 Lund, Sweden; 2Department of Clinical Microbiology, Umeå University, 90185 Umeå, Sweden; 3Department of Medical Microbiology, University Medical Center Utrecht, 3584 CX Utrecht, The Netherlands; 4Department of Laboratory Medicine, Karolinska Institutet, 14152 Huddinge, Sweden; 5Department of Clinical Sciences Lund, Lund University, 22184 Lund, Sweden; 6Department of Translational Medicine, Lund University, 20502 Malmö, Sweden; 7National Laboratory for Public Health, Bissau 1041, Guinea-Bissau; 8Department of Medicine Huddinge, Karolinska Institutet, 14186 Stockholm, Sweden; 9Nuffield Department of Medicine, University of Oxford, Oxford OX3 7BN, UK

**Keywords:** HIV-1, HIV-2, viremia, B-cell phenotype, T-bet, CD95, immune perturbations

## Abstract

Time to AIDS in HIV-2 infection is approximately twice as long compared to in HIV-1 infection. Despite reduced viremia, HIV-2-infected individuals display signs of chronic immune activation. In HIV-1-infected individuals, B-cell hyperactivation is driven by continuous antigen exposure. However, the contribution of viremia to B-cell perturbations in HIV-2-infected individuals remains largely unexplored. Here, we used polychromatic flow cytometry, consensus hierarchical clustering and pseudotime trajectory inference to characterize B-cells in HIV-1- or HIV-2-infected and in HIV seronegative individuals. We observed increased frequencies of clusters containing hyperactivated T-bet^high^CD95^high^CD27^int^ and proliferating T-bet^+^CD95^high^CD27^+^CD71^+^ memory B-cells in viremic HIV-1 (*p* < 0.001 and *p* < 0.001, respectively), viremic HIV-2 (*p* < 0.001 and *p* = 0.014, respectively) and in treatment-naïve aviremic HIV-2 (*p* = 0.004 and *p* = 0.020, respectively)-infected individuals, compared to seronegative individuals. In contrast, these expansions were not observed in successfully treated HIV-1-infected individuals. Finally, pseudotime trajectory inference showed that T-bet-expressing hyperactivated and proliferating memory B-cell populations were located at the terminal end of two trajectories, in both HIV-1 and HIV-2 infections. As the treatment-naïve aviremic HIV-2-infected individuals, but not the successfully ART-treated HIV-1-infected individuals, showed B-cell perturbations, our data suggest that aviremic HIV-2-infected individuals would also benefit from antiretroviral treatment.

## 1. Introduction

Human immunodeficiency virus type 1 (HIV-1) and type 2 (HIV-2) are causative agents of acquired immunodeficiency syndrome (AIDS), but HIV-2 has been shown to be less pathogenic and less transmissible than HIV-1 [1,2,3]. It was previously suggested that HIV-2 infection had limited impact on the survival for many of the infected individuals [4,5,6]. However, we recently reported that HIV-1- and HIV-2-infected individuals not receiving antiretroviral therapy (ART) in fact display similar disease trajectories, albeit with approximately twice as long time to AIDS and HIV-related death in HIV-2-infected individuals, compared to HIV-1 [2]. The underlying mechanism behind this difference in disease progression rate is not fully understood, but it is believed, in part, to be attributed to the finding that HIV-2 infection frequently induces potent and broad cellular and humoral immune responses [7,8,9,10,11,12,13,14,15]. In line with this, the plasma viral load (VL) in HIV-2 infection has been reported to be significantly lower compared to in HIV-1 infection, both at set-point and when adjusted for CD4^+^ T cell levels [16,17,18,19].

However, despite generally lower VL, HIV-2-infected individuals still display signs of chronic immune activation, with similarities to HIV-1-infected individuals [2,20,21,22,23,24,25,26,27]. Moreover, both HIV-1 and HIV-2-driven immune activation has been reported to cause progressive phenotypic alterations of several cell types, including B-, T-, natural killer (NK) and NKT-cells [20,21,22,24,25,26,27]. In HIV-1-infected individuals, an expansion of transitional, activated naïve and memory B-cells, and terminally differentiated plasmablasts, as well as depletion of resting naïve and memory B-cells, has been reported [28]. Similar B-cell perturbations, driven by both HIV-2-specific and non-specific responses, have been shown to induce activation of B-cells, hypergammaglobulinemia, and loss of memory B-cells [21,22,23,25]. Still, the contribution of viremia to B-cell perturbations in HIV-2-infected individuals is largely unexplored.

A subpopulation of the activated B-cells expanded in HIV-1-infected individuals has been shown to express the T-bet transcription factor [29]. In addition to its role in the differentiation of naïve T-cells into Th1 T-cells, T-bet also promotes a Th1-associated response in several other cell types, including B-cells, and thereby broadly orchestrating a Th1-type response [30]. T-bet is an important regulator of antiviral B-cell responses by promoting the differentiation of B-cells into IgG1^+^ or IgG3^+^ memory cells, as well as plasmablasts and plasma cells (PB/PC) [30]. In HIV-1-infected individuals, the T-bet-expressing memory B-cell population has been shown to dominate the Env-specific response [29]. Moreover, HIV-1 viremia has been suggested to drive the T-bet^high^ memory B-cell population [28,29]. In addition, T-bet expression was described to be highest in B-cell populations displaying signs of activation, such as cells with low CD21 expression and high CD95 expression [29]. However, so far profiles of T-bet-expressing B-cells in HIV-2 infection have not been reported.

In this study, we have performed phenotypic characterization of B-cells in HIV-1 and HIV-2-infected individuals. This included in-depth unsupervised consensus hierarchical clustering and pseudotime trajectory inference analyses. By these advanced bioinformatic tools, we have disentangled specific B-cell perturbations present in HIV-2 infection.

## 2. Materials and Methods

### 2.1. Study Participants

The study participants were enrolled from an occupational police cohort of police officers in Guinea-Bissau [31,32], and included HIV seronegative, and treatment-naïve (or sub-optimally ART-treated with VL > 1000 RNA copies/mL plasma) HIV-1- or HIV-2-infected individuals [33]. The HIV-2-infected individuals were further divided into viremic and aviremic individuals, based on the HIV-2 plasma VL quantification limit of 75 RNA copies/mL [12]. In addition, HIV-1-infected individuals on successful ART and with VL < 1000 RNA copies/mL were included from the same cohort.

### 2.2. Sample Collection, HIV Status, and CD4^+^ T-Cell Level Determinations

All blood samples were collected in Guinea-Bissau and shipped to Sweden, with an intact cold chain. Plasma samples were collected using EDTA vacutainer tubes (BD Biosciences, San Jose, CA, USA). Aliquoted plasma was stored at −80 °C until use. Whole blood was collected using Cyto-Chex BCT tubes (Streck, Omaha, NE, USA), where the immunophenotype of lymphocytes is preserved but rendered inadequate for functional analysis [34], and analyzed within 14 days of collection. HIV infection status, absolute CD4^+^ T-cell counts per µL, and percentage of CD4^+^ T-cells of lymphocytes (CD4%) were determined by serology and flow cytometry, as previously described [11,24]. CD4% was used as a marker of disease progression based on previous reports suggesting that CD4% is less variable than absolute CD4^+^ T-cell counts, particularly in settings with high pathogenic burden and comorbidities [35,36,37,38]. Moreover, CD4% has been shown to correlate with markers of T-cell exhaustion in HIV-1 infection [39], and CD4% has been the primary marker for disease progression in previous studies from the Guinea-Bissau police cohort [40].

### 2.3. Plasma HIV-1 and HIV-2 Viral Load Determinations

With minor modifications, HIV-1 and HIV-2 VL was determined using in-house TaqMan-based quantitative reverse transcriptase PCR (qRT-PCR) protocols, as previously described [12]. In brief, RNA was extracted using the miRNeasy micro Kit (Qiagen, Hilden, Germany), and RNA was quantified by qRT-PCR using the Superscript III Platinum One Step qRT-PCR kit (ThermoFisher Scientific, Waltham, MA, USA). The VL quantification limit was 75 RNA copies/mL plasma for HIV-1 and HIV-2, respectively [12].

### 2.4. Plasma IgG1 and IgG3 Quantification

Plasma IgG1 and IgG3 concentrations were determined using the IgG Subclass Human ELISA Kit (ThermoFisher Scientific, Waltham, MA, USA), according to the manufacturer’s instructions. All plasma samples were diluted 1:2500 for both IgG1 and IgG3 quantifications.

### 2.5. Plasma Protein Quantification

Utilizing a proximity extension assay technology, interleukin 12 (IL-12), IL-18, tumor necrosis factor alpha (TNF-α), interferon-γ (IFN-γ), Chemokine (C-X-C motif) ligand 9 (CXCL9) and CXCL10 (also known as IP-10) were quantified in plasma samples using the Olink^®^ Immuno-oncology panel (Olink Proteomics, Uppsala, Sweden) at Olink Proteomics. Protein concentration is reported as normalized protein expression (NPX) levels.

### 2.6. Flow Cytometry

Whole blood samples, stabilized in Cyto-Chex BCT tubes, were stained with a B-cell phenotyping antibody panel, including antibodies specific for 10 extracellular markers (CD19, CD20, CD24, CD27, CD38, CD71, CD95, HLA-DR, CD3 and CD14) and the intracellular T-bet antigen. Clones, fluorochromes and suppliers are listed in Appendix A. In brief, prior to staining the cells with the antibody panel, the whole blood samples were incubated in red blood cell lysis buffer (BD Biosciences, San Jose, CA, USA) at a ratio of 1:6 for 15 min before washing with PBS/FCS (2%). After DNase (6 U/mL) treatment, extracellular antigens were labeled in PBS/EDTA (2 mM) for 30 min. Cells were thereafter permeabilized using the FoxP3 kit (eBiosciences, San Diego, CA, USA) prior to intracellular staining for 30 min. The cells were washed and resuspended in Cytofix Buffer (BD Biosciences, San Jose, CA, USA) prior to analysis on a BD Fortessa instrument (BD Biosciences, San Jose, CA, USA).

### 2.7. Data Processing and Statistical Analysis

For the investigation of CD95, T-bet, and CD27 expression on activated B-cells, single CD14^−^CD3^−^CD19^+^CD20^+^ cells were obtained by manual gating using the FlowJo™ v10.8 Software (BD Life Sciences, San Jose, CA, USA) (Appendix A). Compensated values from single B-cells (CD14^-^CD3^−^CD19^+^) were exported using the FlowJo™ v10.8 Software (Appendix A). Exported data was further pre-processed and clustered using the FlowSOM algorithm in RStudio (v1.4.1717) as previously described [41,42,43]. In short, cells from all samples were loaded (n = 5,130,630 cells), the data was logicle transformed using the flowCore package (v2.4.0) and residual doublets were removed using the PeacoQC package (v1.3.3) [44,45]. A FlowSOM object was computed using the FlowSOM package (v2.0.0), with settings specifying a 10 × 10 grid and rlen = 100, based on the expression of CD24, HLA-DR, CD71, CD27, T-bet, CD38, CD95 and CD20. We applied maxMeta = 45 to allow the FlowSOM function to specify the number of meta-clusters. The principal component analysis of FlowSOM cluster frequency was performed using the built-in R function prcomp() [46]. Difference in marker expression levels between FlowSOM clusters was determined based on the relative difference in expression of each marker.

The Uniform Manifold Approximation and Projection (UMAP) was based on the expression of the above-mentioned markers from 50,000 cells per HIV status group, with equal sampling from each sample within each infection group, using the uwot package (v0.1.11) [47]. The same subsampled cells were used for subsequent pseudotime trajectory inference analysis using the Slingshot package (v2.0.0), as previously described [48,49]. We specified approx_points = 100 to reduce the computational time due to the large size of our data set [49]. The ggplot2 R package (v3.3.5) [50] was used to produce both UMAP and density plots, to display marker expression in each FlowSOM cluster, density of PC1 values of individuals within each HIV status group, and density of cells from each HIV status group by pseudotime in each pseudotime trajectory.

Differences in frequencies in manually gated B-cell subpopulations and FlowSOM generated clusters, as well as in principal component one (PC1) values, were calculated using the non-parametric Kruskal–Wallis test with Dunn’s post hoc test comparing all HIV status groups, and Mann–Whitney U test for comparisons of two groups, using GraphPad Prism version 9.2.0 (GraphPad Software, Inc., La Jolla, CA, USA). Spearman rank correlation analysis was performed to test for correlations between cluster frequencies and CD4%, VL and plasma proteins concentrations.

## 3. Results

### 3.1. Study Participants

To characterize B-cell phenotypes in HIV-1 and HIV-2 infections, blood samples from HIV-1 (n = 15) and HIV-2 (n = 20)-infected, and HIV seronegative (n = 25) individuals were analyzed (Table 1) [31,32]. The HIV-1-infected participants were either treatment naïve or sub-optimally treated (viremic HIV-1; n = 8), or successfully treated (ART-treated HIV-1; n = 7). The HIV-2-infected individuals, either treatment naïve (n = 19) or sub-optimally treated (n = 1), could be further divided into viremic (n = 8) and aviremic individuals (n = 12) based on the level of quantifiable HIV-2 plasma VL, above or below the quantification limit of 75 RNA copies/mL plasma. Both CD4% and absolute CD4^+^ T-cell counts were significantly lower in viremic HIV-1 (*p* < 0.001 and *p* < 0.001, respectively) and viremic HIV-2-infected individuals (*p* < 0.001 and *p* = 0.001, respectively), compared to HIV seronegative individuals. Additional characteristics of the study participants are described in Table 1.

### 3.2. Both HIV-1 and HIV-2 Infection Induce an Expansion of T-Bet and CD95-Expressing B-Cells

To study the impact of HIV-1 and HIV-2 infection on B-cell activation, we investigated the expression of the activation marker CD95 and the transcription factor T-bet, by CD19^+^CD20^+^ B-cells in HIV-1- or HIV-2-infected, and HIV seronegative individuals. A clear skewing of the T-bet and CD95 expression patterns was observed in all treatment-naïve or sub-optimally treated HIV-1- or HIV-2-infected individuals, as compared to the HIV seronegative individuals (Figure 1A,B). In contrast, studied B-cell populations were not found to be altered in HIV-1-infected individuals on ART. The majority of the T-bet^+^CD95^+^ B-cells were CD27^−^ (Figure 1C). Increased frequencies of T-bet^+^CD95^+^CD27^−^ B-cells were observed in viremic HIV-1 (*p* < 0.001), viremic HIV-2 (*p* < 0.001), and treatment-naïve aviremic HIV-2 (*p* = 0.030)-infected individuals, compared with HIV seronegative individuals (Figure 1D). In addition, a significant expansion of T-bet^+^CD95^+^CD27^+^ B-cells was observed in viremic HIV-1 (*p* < 0.001), and viremic HIV-2 (*p* = 0.002)-infected individuals, but not in aviremic HIV-2-infected individuals, compared to HIV seronegative individuals (Figure 1D).

Taken together, these findings show that both HIV-1 and HIV-2 infection induces an expansion of activated T-bet-expressing B-cells.

### 3.3. Hierarchical Clustering Analysis Shows That HIV-1 and HIV-2 Infections Induce Phenotypic Perturbations in the B-Cell Compartment

To investigate the impact of HIV-1 and HIV-2 viremia on B-cell perturbations, we performed unsupervised consensus hierarchical clustering using the FlowSOM algorithm [43]. The analysis suggested 16 distinct clusters of B-cells, based on the expression of eight assessed markers (Appendix A). Following quality control of each cluster, where the expression pattern of analyzed markers was compared to previously reported B-cell populations [51], clusters 13–16 were excluded from further analyses on the rationale of minor size and surface marker expression patterns that do not correspond to previously described B-cell populations (Appendix A). The remaining 12 clusters could be divided into four cluster groups; transitional/naïve-like B-cells (cluster 1–2), memory-like B-cells (Cluster 3–7), T-bet^+/high^ memory-like B-cells (cluster 8–11), and PB/PCs (cluster 12) (Figure 2A). The transitional/naïve-like B-cell cluster group was characterized by clusters of cells expressing high to intermediate CD38 levels and no CD27 (Figure 2B,C). CD24 expression was highest in cluster 1 (transitional B-cells) and low in cluster 2 (naïve-like B-cells), in accordance with previous B-cell characterization [51]. The memory-like B-cell cluster group (cluster 3–7) contained resting and activated memory B-cells expressing CD27 and intermediate to high levels of CD95. Cluster 3 contained activated naïve-like/early memory B-cells with intermediate CD27, CD24 and CD95 expression, but with CD38 expression at levels comparable to naïve-like B-cells in cluster 2. In contrast, the B-cells in cluster 4 had downregulated CD38, a sign of differentiation into memory cells [52], despite lower CD27 expression levels compared to cluster 3 B-cells. Cluster 5 represented resting memory cells with high CD27 and low CD38 expression, while cluster 6 and 7 were activated memory cells expressing CD27, CD38 and CD95. The T-bet^+/high^ memory-like B-cell cluster group (clusters 8–11) contained activated memory B-cells expressing low to high levels of CD27, intermediate to high levels of CD95 and T-bet. The PB/PC cluster group (clusters 12) included B-cells negative for CD20, with high CD27 and CD38, and low to intermediate HLA-DR, expression levels.

A principal component analysis performed on FlowSOM cluster frequency showed statistically significant differences in PC1 between viremic HIV-1 (*p* < 0.001), viremic HIV-2 (*p* = 0.001), and treatment-naïve aviremic HIV-2-infected individuals (*p* = 0.013), compared to HIV seronegative individuals (Figure 2D). PC1 in ART-treated HIV-1-infected individuals did not separate from HIV seronegative, but it was significantly different from viremic HIV-1-infected individuals (*p* = 0.026). The PC1 value correlated with increasing CD4% (r = 0.695; *p* < 0.001) and decreasing VL (r = −0.560; *p* = 0.002) when all ART-naïve or sub-optimally treated HIV-1- and HIV-2-infected individuals were analyzed together. The significant correlation with CD4% remained when only HIV-2-infected individuals were analyzed (r = 0.477; *p* = 0.034).

In summary, the unsupervised consensus hierarchical clustering identified B-cell perturbations in individuals with HIV-1 or HIV-2 infection.

### 3.4. T-Bet-Expressing Hyperactivated B-Cells, Identified by Hierarchical Clustering Analysis, Distinguish Aviremic HIV-2-Infected from Seronegative Individuals

To further dissect phenotypic alterations in the B-cell compartment caused by HIV-1 and HIV-2 infections, we compared frequencies of the 12 FlowSOM clusters identified in each HIV status group (Figure 3A). The frequency of cluster 8, 9 and 11, representing T-bet^high^CD95^high^CD27^int^ hyperactivated memory cells, T-bet^+^CD95^high^CD71^+^ proliferating T-bet^+^ memory cells and T-bet^high^CD95^int^CD27^−^ hyperactivated memory cells, were significantly higher in viremic HIV-1 (*p* < 0.001, *p* < 0.001, and *p* = 0.009, respectively) and viremic HIV-2 (*p* < 0.001, *p* = 0.014, and *p* < 0.001, respectively)-infected individuals compared to HIV seronegative individuals. Moreover, the frequency of cluster 2, containing resting naïve-like cells, was lower in viremic HIV-1 (*p* < 0.001), and viremic HIV-2 (*p* = 0.004)-infected individuals, compared to HIV seronegative individuals (Figure 3B and Appendix A). Successful ART treatment of HIV-1-infected individuals reversed the depletion of cluster 2 (*p* = 0.017). Treatment-naïve aviremic HIV-2-infected individuals separated from HIV seronegative individuals by higher frequencies of cluster 6, consisting of T-bet^int^CD95^+^ activated memory cells, (*p* = 0.010), cluster 8 (*p* = 0.004) and cluster 9 (*p* = 0.020), as well as lower frequencies of cluster 2 (*p* = 0.019). Of note, cluster 8 and 11 frequency distinguished viremic from aviremic HIV-2-infected individuals (*p* = 0.001 and *p* = 0.004, respectively) in direct comparisons, but this difference was not significant in the multiple comparison containing all HIV status groups. Since T-bet promotes class-switching to IgG1 and IgG3 [30], we next investigated the correlation between the frequency of the T-bet^high^-expressing clusters, 8 and 11, and IgG1 or IgG3 plasma titers. The results showed that both IgG1 and IgG3 titers correlated with the T-bet^high^-expressing B-cells, i.e., cluster 8 (r = 0.588; *p* = 0.023, and r = 0.625; *p* = 0.015, respectively) and cluster 11 (r = 0.781; *p* < 0.001, and r = 0.618; *p* = 0.016, respectively) frequencies among the HIV-1-infected individuals (Appendix A). IgG1 titers also correlated with the cluster 8 (r = 0.446; *p* = 0.048) frequency among the HIV-2-infected individuals (Appendix A). Furthermore, the frequency of cluster 8 in HIV-2-infected individuals was associated with plasma VL (r = 0.704; *p* < 0.001), and CD4% (r = 0.723; *p* < 0.001), in addition to plasma concentrations of proinflammatory Th1-associated cytokines: IL-12 (r = 0.549; *p* = 0.015), IL-18 (r = 0.468; *p* = 0.043), and TNF-α (r = 0.686; *p* = 0.001), IFN-γ (r = 0.530; *p* = 0.020); and chemokines CXCL9 (r = 0.619; *p* = 0.005), and CXCL10 (r = 674; *p* = 0.002) (Appendix A).

Intriguingly, the frequency of cluster 3, containing CD95^int^CD27^int^ activated naïve-like/early memory B-cells, was found to be significantly higher in viremic and aviremic HIV-2-infected individuals compared to HIV seronegative individuals (*p* = 0.002, *p* = 0.009, respectively, Appendix A). However, since the frequency of naïve-like B-cells (i.e., cluster 2) was shown to be reduced in viremic HIV-1-infected individuals, we hypothesized that the reason for not observing any increase in activated naïve-like/early memory B-cells in this group was due to the low abundance of naïve-like B-cells. We therefore determined the ratio of activated naïve-like/early memory B-cells (cluster 3) to CD95^-^CD27^-^ resting naïve-like (cluster 2) B-cells. The analysis showed that the ratio of activated vs. resting naïve-like B-cells was significantly higher in viremic HIV-1 (*p* < 0.001), viremic HIV-2 (*p* < 0.001), and aviremic HIV-2 (*p* = 0.003)-infected individuals compared with HIV seronegative individuals (Appendix A).

Unsupervised hierarchical clustering analysis showed that, independent of viremia, especially hyperactivated T-bet-expressing memory B-cells were elevated, whereas naïve-like B-cells were reduced, in treatment-naïve or sub-optimally treated HIV-1- and HIV-2-infected individuals.

### 3.5. Both HIV-1 and HIV-2 Infection Promotes B-Cell Differentiation Accompanied by T-Bet Expression

As an increase in frequencies of B-cell clusters containing T-bet^high^ activated memory cells was observed, we used the Slingshot method [48] to infer pseudotime trajectories for the purpose of investigating the impact of HIV-1 and HIV-2 infections on B-cell differentiation. Cluster 1, containing CD27^−^CD24^high^CD38^high^ transitional B-cells, was designated as the starting population since this was the least differentiated B-cell subtype detected [52]. The analysis indicated four distinct trajectories, where clusters 1-6 represented the root of all lineages (Figure 4A,B). The trajectory analysis subsequently supported branching of trajectory 1 into CD20^-^HLA-DR^int/−^ PB/PCs (cluster 12), trajectory 2 into T-bet^high^CD38^low^ activated memory B cells (cluster 10), trajectory 3 into T-bet^high^CD27^int^ hyperactivated memory B-cells (cluster 8), and trajectory 4 into T-bet^+^CD71^+^ proliferating memory B-cells (cluster 9). To identify the impact of HIV-1 and HIV-2 infections and viremia on B-cell differentiation, each pseudotime trajectory was stratified by respective HIV status group (Figure 4C and Appendix A). While B-cells from HIV seronegative individuals were found throughout the entire trajectory of lineage 1, B-cells at the terminal ends of trajectory 2–4 were predominantly from HIV-1- and HIV-2-infected individuals (Figure 4C). This was especially clear in trajectory 3 and 4, where the clusters at the terminal end included cluster 8 and 9, respectively. These clusters were previously shown to have significantly higher frequencies in viremic HIV-1-, viremic HIV-2- and aviremic HIV-2-infected individuals compared to HIV seronegative individuals (Figure 3B). In contrast, no clear enrichment of B-cells at the terminal ends of lineage 3 and 4 was observed among HIV-1-infected individuals receiving ART. Thus, the pseudotime trajectory inference analysis showed skewing of B-cell differentiation towards activation accompanied with increased T-bet expression in both HIV-1 and HIV-2 infection.

## 4. Discussion

In this study, we show that both HIV-1 and HIV-2 infection induce an expansion of hyperactivated T-bet-expressing B-cells. In contrast to successfully treated HIV-1-infected individuals, treatment-naïve aviremic HIV-2-infected individuals could be distinguished from seronegative individuals by increased frequency of T-bet-expressing B-cells, supported by both a manual gating strategy and a consensus hierarchical clustering approach. This suggests that HIV-2 infection perturbs the phenotype of B-cells despite unquantifiable viremia.

T-bet has previously been described to be upregulated in several cell types following viral infection as part of a multi-cell type Th1-associated antiviral response [28,53]. Knox et al. reported that T-bet expression was highest in activated memory cells [29], defined as CD27^+^CD21^−^ memory and CD27^−^CD21^−^ tissue-like memory (TLM) cells. However, despite being heterogenous populations, in depth characterization of T-bet expression in distinct subpopulations was not attempted as in the current study. The association between T-bet^+^ memory B-cell populations and pathogenic outcome, for example following rhinovirus exposure and kidney transplantation, has previously been studied using unsupervised cluster analysis [54,55]. We therefore choose to apply hierarchical cluster analysis to characterize distinct T-bet-expressing B-cell subpopulations in HIV-1 and HIV-2 infections. Of note, cluster 8 and 9 (which distinguished both viremic and aviremic HIV-2-infected individuals from seronegative controls) contained T-bet^high^CD27^int/+^ hyperactivated memory B-cells and T-bet^+^CD95^high^CD71^+^ proliferating T-bet^+^ memory cells, respectively. This likely represent a mixture of the two activated memory populations described by Knox et al. [29]. Moreover, HIV-1 infection has previously been shown to induce the expansion of a CD20^high^CD27^int^ hyperactivated memory B-cell population [56] at similar frequencies compared to the frequency of cluster 8 observed in HIV-1-infected individuals in our study. Cluster 9, which contained CD71^+^ proliferating memory B-cells, most likely instead resembles the B-cell population found to be expanded following Ebola and Influenza infections, as observed by Ellebedy et al. [57]. Thus, based on the immunophenotypic similarity, cluster 9 could represent either re-activated memory B-cells, or a novel lineage of memory B-cell arising from activated naïve B-cells, as a part of an ongoing anti-viral response. However, and as previously suggested, sequential B-cell receptor (BCR) clonal analysis would be required to delineate the origin of these cells and to determine if they further differentiate into antibody secreting cells or re-enter the pool of resting memory B-cells [57]. In addition, our pseudotime trajectory inference analyses showed that cluster 9 was located at the terminal end of lineage 4, which supports the hypothesis that these cells would not further differentiate into antibody secreting cells, but rather become resting memory B-cells.

Although aviremic HIV-2-infected participants in our study tended to harbor lower frequencies of cluster 8 compared to viremic HIV-2-infected individuals, the aviremic HIV-2-infected individuals still had significantly higher frequencies of cluster 8 compared to HIV seronegative individuals. In contrast, this was not observed among successfully treated HIV-1-infected individuals. In accordance with this, HIV-1-induced expansion of T-bet^high^ memory B-cells has previously not been observed at higher frequencies in successfully treated HIV-1-infected individuals [29]. This suggests that, in contrast to treatment suppressed HIV-1 infection, HIV-2 infection promotes the expansion and activation of T-bet-expressing memory cells, even in the absence of quantifiable plasma viremia. These observations are also in line with our previous findings showing that the frequency of T-bet^+^ CD4^+^ T-cells, expressing activation and exhaustion markers, are higher in aviremic HIV-2-infected individuals compared to HIV seronegative individuals [20]. Moreover, a recent phenotypic characterization of CD8^+^ T-cells, in study participants also included in the current study, showed that increased frequencies of activated and exhausted CD8^+^ T-cells distinguish aviremic HIV-2-infected individuals from HIV seronegative individuals [24]. In line with the findings of the current study on hyperactivated B-cells, the frequency of activated and exhausted CD8^+^ T-cells in HIV-2-infected individuals correlated with plasma levels of the inflammation marker CXCL10, and also soluble CD14 and beta-2 microglobulin [24]. These plasma inflammation markers have also recently been shown by us to correlate with expression levels of interferon alpha-inducible protein 27 (IFI27) in HIV-2-infected individuals [58]. Furthermore, the expression of IFI27 was shown to distinguish aviremic HIV-2-infected from HIV seronegative individuals [58]. Taken together, this suggests that inflammation and immune activation, including responses within Type I and II IFN signaling pathways, are elevated among aviremic HIV-2-infected compared to HIV seronegative individuals.

We also observed an increased proportion of activated naïve-like/early memory B-cells. HIV-1 infection-induced expansion of activated naïve B-cells has been suggested to be viremia-dependent, as elevated frequencies of CD95^+^ activated naïve B-cells were observed in viremic HIV-1-infected individuals, but not in HIV-1-infected individuals receiving successful ART treatment [59]. However, the significantly higher cluster ratio of activated naïve-like/early memory B-cells, identified in cluster 3, to resting naïve-like B-cells, in cluster 2, in aviremic HIV-2-infected individuals compared to HIV seronegative individuals, suggest increased stimulation of naïve B-cells despite unquantifiable HIV-2 viremia. Possible explanations for such immune activation, despite low or no HIV-2 viremia, could be virus replication in other compartments or long duration of the HIV-2 infection. Indeed, the median time from estimated infection or diagnosis was more than 17 years for our study participants (data not shown). It is still not known whether active HIV-2 replication occurs in aviremic HIV-2-infected individuals, or not. However, intracellular viral mRNA has been detected in peripheral blood mononuclear cells of aviremic HIV-2-infected individuals [60,61]. Similarly, both Gag and viral mRNA has been detected in sigmoid and ileum biopsies from aviremic HIV-2-infected individuals [62]. Moreover, Fumarola et al. recently reported that CD4% recovered in HIV-2-infected individuals receiving ART, while untreated HIV-2-infected individuals displayed declining CD4% [63], further supporting ongoing virus replication in aviremic HIV-2-infected individuals.

The effect of HIV-2 infection on B-cells is much less studied than the effect of HIV-1 infection, but both HIV-1 and HIV-2 infections have been reported to induce a depletion of isotype switched and unswitched memory B-cells [21,23,25]. In HIV-1-infected individuals, this depletion has been shown to occur with a concomitant expansion of activated TLM B-cells [56]. Although the frequency of TLM B-cells have not been studied in HIV-2-infected individuals, two studies report on immunophenotypically similar subpopulations. Ponnan et al. observed an increase in atypical, CD27^-^IgD^-^, memory B-cell frequency in HIV-2-infected individuals receiving ART [22], but this has not been studied in ART-naïve HIV-2-infected individuals yet. Honge et al. reported an increased frequency of CD21^low^ B-cells in both viremic and aviremic HIV-2-infected individuals [21], suggesting that TLM B-cells are indeed expanded in HIV-2-infected individuals as well. In line with this, our principal component analysis of FlowSOM cluster frequency indicated HIV-2 infection-induced perturbations of the B-cell compartment driven by the expansions of activated B-cell populations and depletion of resting B-cell populations. In addition, we observed an HIV-induced expansion of cluster 8, 9 and 11, likely containing a mixture of both activated memory and TLM B-cells.

Due to limited availability of clinical samples from the study participants, we were not able to differentiate phenotypes of HIV-specific and non-HIV-specific B-cells. However, Knox et al. has previously reported that HIV-1 gp140-specific B-cells represents approximately 1% of all class-switched memory B-cells [29]. As the frequency of the T-bet^high^ clusters far exceeds this, the perturbations observed in the current study should involve non-HIV-specific activation of B-cells. In line with this, we observed that the frequency of cluster 8 correlated with hypergammaglobulinemia and increasing plasma concentrations of Th1-associated proinflammatory cytokines and chemokines in HIV-2-infected individuals, with the strongest correlations to TNF-α and CXCL10. Interestingly, Zumaquero et al. has previously reported that the expansion of T-bet^high^ B-cells in patients with active SLE, i.e., chronic autoantigen stimulation, correlated with CXCL10 serum concentration [64].

In this study, we utilized both unsupervised consensus hierarchical clustering and pseudotime trajectory inference analysis to study the impact of HIV infection on the B-cell phenotypes. As most pseudotime trajectory inference analysis tools often are used to analyze a small number of cells, usually in the range of tens of thousands of cells, these tools have so far mainly been used to analyze single-cell RNA sequencing data rather than flow cytometry data [65]. However, recent developments have shown the capacity of analyzing hundreds of thousands of cells [49,66]. In a comparison of 45 different trajectory inference methods, with regard to accuracy, stability, scalability, and usability of each method, Slingshot was found to perform well across all four criteria [65]. Moreover, Slingshot was previously shown to produce pseudotime trajectories resembling known CD8^+^ T-cell differentiation trajectories using flow cytometry data acquired with the same number of markers used here. Thus, we chose to perform the pseudotime trajectory inference analysis using the Slingshot algorithm [48,49], which supported the definition of four separate B-cell differentiation lineages. The observed trajectories were in line with previously described B-cell differentiation patterns [52]. Lineage 1 was the only lineage where cells from HIV seronegative donors were detected throughout the full spectrum of the pseudotime trajectory. The remaining lineages branched away from the PB/PC trajectory and instead differentiated into T-bet^high^/T-bet^+^ memory B-cells. As described above, lineage 4 contained B-cells progressing from transitional B-cells to proliferating memory B-cells, which according to Ellebedy et al. could represent clonally expanded memory B-cells that would eventually become resting memory B-cells [57]. However, since Slingshot can only detect linear trajectories such circular differentiation pathways would not be detected. Furthermore, sequential BCR sequencing of the memory B-cell population would be required to prove if these cells indeed become resting memory B-cells. Within lineage 3, B-cells differentiated from transitional B-cells into hyperactivated T-bet^high^ memory B-cells (cluster 9). Interestingly, cluster 9 cells were also part of lineage 4 where they eventually differentiated into proliferating B-cells represented within cluster 10. Longitudinal studies would be required to improve our understanding of T-bet^+^ memory B-cell fate, since they have been suggested to be able to give rise to PB/PCs and/or self-renew [30].

In conclusion, independent of plasma viremia, HIV-2-infected participants had increased frequencies of both activated naïve, and hyperactivated memory B-cells. These observations display, for the first time, the maintenance of T-bet^high^ B-cell populations despite unquantifiable plasma VL among HIV-infected individuals. Together with our previous findings that the majority of HIV-2-infected individuals will develop AIDS [2], it is likely that aviremic HIV-2-infected individuals would also benefit from early ART initiation, for the purpose of inhibiting B-cell perturbations.

## Figures and Tables

**Figure 1 cells-11-03142-f001:**
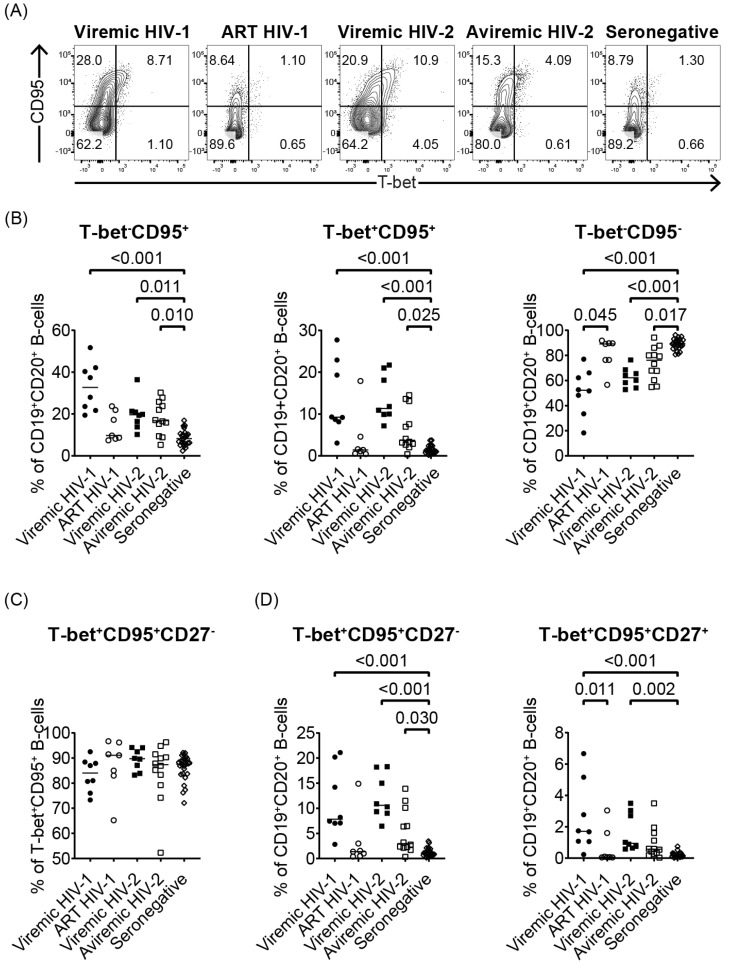
Both HIV-1 and HIV-2 infection induces T-bet expression in CD19^+^CD20^+^ B-cells. (**A**) Representative flow cytometry plots depicting CD95 and T-bet expression on CD19^+^CD20^+^ B-cells in treatment-naïve or sub-optimally ART-treated HIV-1 (viremic HIV-1, n=8), successfully ART-treated HIV-1 (ART HIV-1, n = 7), viremic HIV-2 (viremic HIV-2, n = 8), treatment-naïve aviremic HIV-2 (aviremic HIV-2, n = 12)-infected individuals and HIV seronegative (seronegative, n = 25) individuals. (**B**) Scatter plots illustrating the percentage of T-bet^+^CD95^+^, T-bet^+^CD95^+^ and T-bet^+^CD95^+^ among B-cells in the different study groups. (**C**) Scatter plots illustrating the percentage of T-bet^+^CD95^+^CD27^-^ among T-bet^+^CD95^+^ B-cells (CD19^+^CD20^+^). (**D**) Scatter plots illustrating percentage of T-bet^+^CD95^+^CD27^-^ and T-bet^+^CD95^+^CD27^+^ cells among CD19^+^CD20^+^ B-cells. Nonparametric Kruskal–Wallis test followed by Dunn’s post hoc was performed to compare groups. Medians are depicted in scatter plots.

**Figure 2 cells-11-03142-f002:**
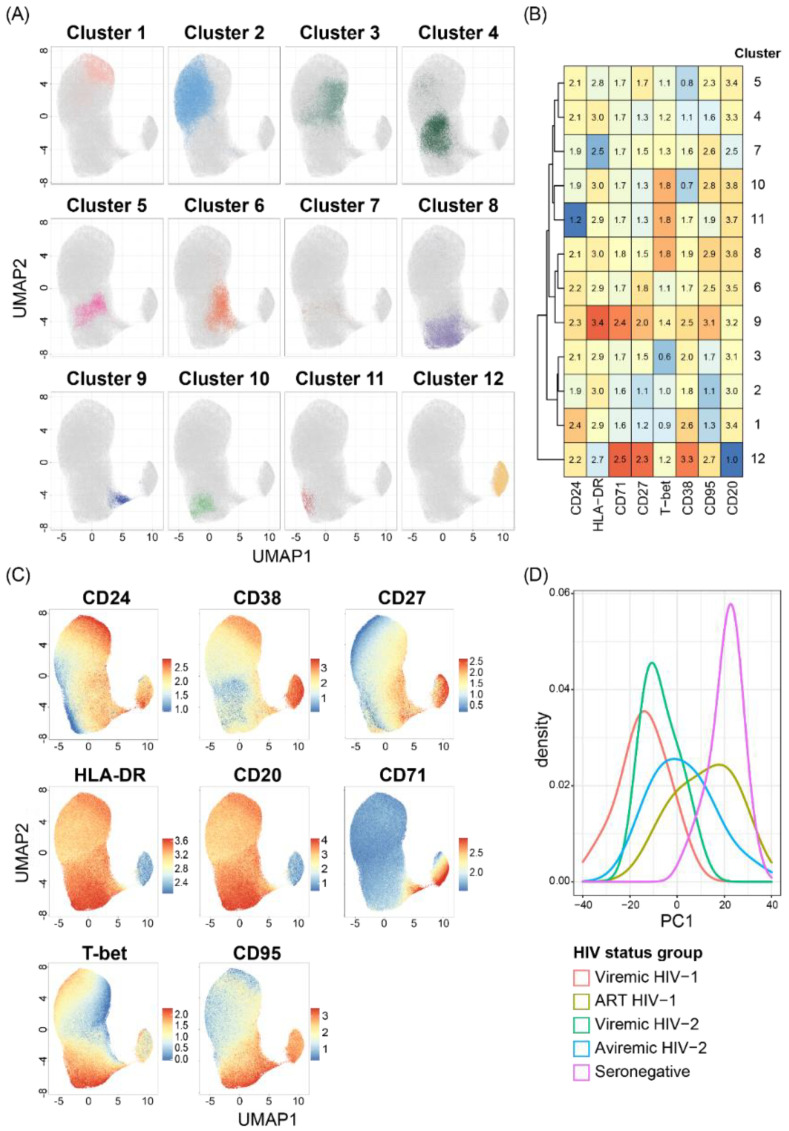
FlowSOM analysis showed alterations in B-cell phenotypes among HIV-1- and HIV-2-infected individuals. B-cells from all study participants were clustered according to CD24, CD38, CD27, HLA-DR, CD20, CD71, T-bet and CD95 expression using the FlowSOM algorithm. (**A**) FlowSOM clusters projected on a UMAP plot, generated using an equal number of cells from each HIV status group, based on the expression of the above mentioned eight markers. (**B**) Heatmap displaying scaled marker expression values within each FlowSOM cluster, and the color coding is based on marker expression centered and scaled by column (**C**) Expression of the eight markers is visualized on individual UMAP plots. (**D**) A principal component analysis (PCA) was performed on FlowSOM cluster frequency. The density plot display PC1 values, for treatment-naïve or sub-optimally ART-treated HIV-1 (viremic HIV-1, n = 8), successfully ART-treated HIV-1 (ART HIV-1, n = 7), viremic HIV-2 (viremic HIV-2, n = 8), treatment-naïve aviremic HIV-2 (aviremic HIV-2, n = 12)-infected individuals and HIV seronegative (seronegative, n = 25) individuals.

**Figure 3 cells-11-03142-f003:**
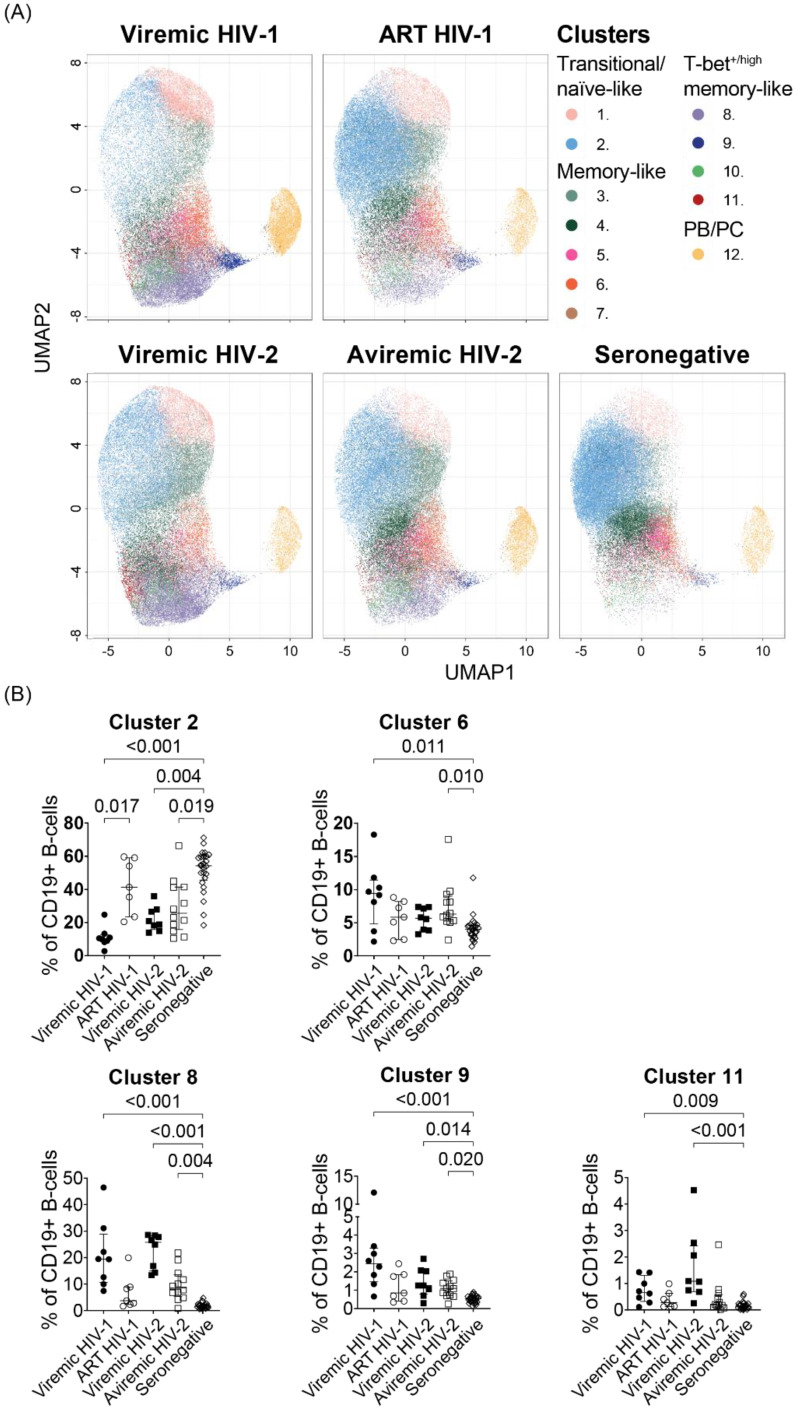
Specific B-cell perturbations within HIV-1- and HIV-2-infected individuals were identified by consensus hierarchical clustering. (**A**) FlowSOM clusters projected on UMAP plots faceted by HIV status group. (**B**) FlowSOM cluster frequency in treatment-naïve or sub-optimally ART-treated HIV-1 (viremic HIV-1, n = 8), successfully ART-treated HIV-1 (ART HIV-1, n = 7), viremic HIV-2 (viremic HIV-2, n = 8), treatment-naïve aviremic HIV-2 (aviremic HIV-2, n = 12)-infected individuals and HIV seronegative (seronegative, n = 25) individuals. Nonparametric Kruskal–Wallis test followed by Dunn’s post hoc was performed to compare groups. Medians and IQR are depicted in scatter plots.

**Figure 4 cells-11-03142-f004:**
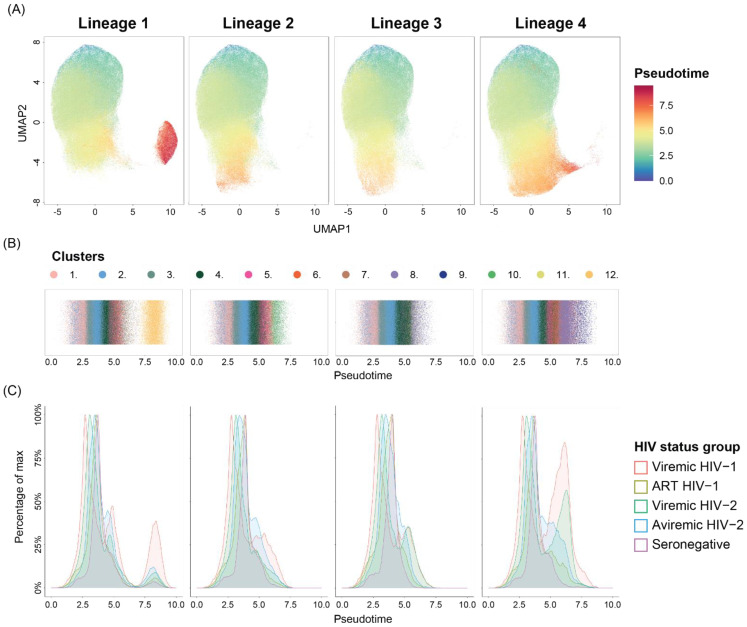
HIV-1 and HIV-2 infection promotes differentiation of B-cells into T-bet^high^ activated memory B-cells. (**A**) Pseudotime projected on UMAP plots faceted by identified lineage. The pseudotime trajectory inference analysis, based on FlowSOM clusters, was performed using the Slingshot method. Fifty thousand cells from each HIV status group were included in the analysis. (**B**) Jitterplots colored by FlowSOM clusters along pseudotime in each lineage. (**C**) Lineage faceted density plots displaying the distribution of cells along pseudotime, from treatment-naïve or sub-optimally ART-treated HIV-1 (viremic HIV-1, n = 8), successfully ART-treated HIV-1 (ART HIV-1, n = 7), viremic HIV-2 (viremic HIV-2, n = 8), aviremic HIV-2 (aviremic HIV-2, n = 12)-infected individuals and HIV seronegative (seronegative, n = 25) individuals.

**Table 1 cells-11-03142-t001:** Characteristics of study participants ^a^.

Characteristic	Viremic HIV-1 ^b^	ART HIV-1 ^c^	Viremic HIV-2 ^d^	Aviremic HIV-2 ^e^	HIV Seronegative
Numbers (female/male)	8 (3/5)	7 (3/4)	8 (0/8)	12 (3/9)	25 (9/16)
Age in years ^f^	46 (42–50)	45 (42–55)	57 (55–60)	58 (51–60)	56 (51–61)
% CD4^+^ T-cells of lymphocytes ^f^	10 (6–12)	32 (18–36)	14 (13–15)	33 (25–36)	39 (38–45)
CD4^+^ T-cells (cells/µL) ^f^	239 (124–321)	537 (322–767)	267 (205–440)	557 (416–957)	924 (859–1204)
Viral load (copies/mm^3^) ^f,g^	35,963 (24,652–50,144)	<75 (<75)	2966 (1048–7857)	<75	NA

^a^ Characteristics of study participants (n = 60) grouped by HIV status and treatment. ^b^ HIV-1-infected individuals either treatment naïve or receiving suboptimal ART (viremic HIV-1, plasma VL > 1000 RNA copies/mL); ^c^ HIV-1-infected individuals receiving ART with viral control (ART HIV-1, plasma VL < 1000 RNA copies/mL); ^d^ HIV-2-infected viremic individuals, either treatment naïve (VL > 75 RNA copies/mL) or receiving suboptimal ART (VL > 1000 RNA copies/mL); ^e^ HIV-2-infected aviremic individuals naïve to treatment (plasma VL < 75 RNA copies/mL); ^f^ Median (interquartile range (IQR)); ^g^ Viral load quantification limit was 75 copies/mL plasma for both HIV-1 and HIV-2; NA = not applicable.

## Data Availability

All raw data supporting the conclusions of this article are available upon request.

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
