# Peer review of "Hierarchical Clustering and Trajectory Analyses Reveal Viremia-Independent B-Cell Perturbations in HIV-2 Infection"

_cells, 2022, doi:10.3390/cells11193142_

Round 1

Reviewer 1 Report

In this manuscript, Johansson et al. compared B-cell populations from the blood of HIV-1 viremic, HIV ART-successful, HIV-2 viremic, HIV-2 aviremic and HIV negative donors, using hierarchical clustering. Their data indicate that viremic individuals have elevated levels of T-bet+ hyperactivated and proliferating B-cells. Additionally, HIV-2 aviremic individuals showed evidence of elevated naïve B-cell stimulation and other alterations in B-cell populations that the authors interpret as an ongoing viral response. The manuscript is well-written and provides new incite into changes in B-cell populations associated with HIV-2 infection.

The limitation of this study was the small number of antibodies used for phenotyping. HIV infection is reported to be associated with atypical B-cell phenotypes and B-cell exhaustion; CD21, IgD, CD10 and FCRL4 would have been useful. The small antibody panel made it difficult to distinguish the FlowSOM-generated clusters. For better visualization of figure 3A, the authors might consider including a set of supplementary figures in which an individual clusters of interest retain their color-shading while the others are light-gray-shaded.

In the figures, it would be extremely helpful to include the phenotype and proposed B-cell type with the cluster number.

A legend for Fig. S3 would be useful.

Is the reference defined/described in the manuscript that was used to define/gauge antibody-binding (Fig. S2)?

It is apparent that cluster 8 is T-bet high but not that cluster 9 is (abstract defines it as +; manuscript body as high). The scale protein expression in Fig. 2B shows a protein expression value of 1.4 for cluster 9 T-bet. Cluster 8 has 1.5 for CD27, which the authors define as CD27int. Please clarify how expression is defined as high and intermediate.

While clusters associated aviremic HIV-2 compared to those for seronegative suggest an ongoing viral response, clusters associated with aviremic HIV-2 are similar to those for ART HIV-1. Supposedly individuals from this cohort have been on ART for a while. Therefore, the remaining provirus in these individuals is transcriptionally silent. It is therefore possible that the clusters associated with aviremic HIV-2 do not represent an ongoing viral response.

Other than T-cell decline, have there been other studies analyzing cytokine profiles of T-cell function to complement this study suggesting an ongoing antiviral response in HIV-2 aviremic individuals?

Fig. 1B, CD27+ should be CD27-.

Reviewer 2 Report

Johansson et al. characterize in detail B-cells in cART-treated HIV-1-infected patients, both viremic and aviremic, in HIV-2-infected patients, both viremic and naïve aviremic, and in HIV-seronegative individuals.

Analyses are well described and executed, and the study provides a thorough profiling of the characteristics of HIV-1 and HIV-2 impact on B-cell perturbations by using bioinformatic tools and trajectory analyses.

The Authors demonstrate the hyperactivation of memory B-cells, showing high T-bet expression in HIV-1- and HIV-2-infected individuals, which was found to be viremia-independent. Based on the data obtained, this report could be valuable for understanding how HIV-1 and HIV-2 perturb certain B-cell subpopulations.

This is an interesting and worth publishing work, however some minor points should be addressed as below.

1.    In the Results section, page 5 line 200, the Authors describe the majority of the T-bet+CD95+-B-cells as CD27-, but in the Figure legend they indicate the same cell population as CD27+, please uniform.

2.    Please consider to make Figure 1 legend clearer.

3.    In the graphs of Figure 1A and 1C, B-cells are indicated as “cells” while in the graph of Figure 1B B-cells are indicated as “B-cells”, please uniform. 

4.    Overall typos mistake control needs to be done, i.e. at page 6 line 215 the dot is in the wrong position, as well as at page 15 line 494.

5.    Please spell out abbreviation at the first time of mention, i.e. PC1 (page 4 line 164).

6.    Page 9 line 301, the Authors mention Figure 3C which is not reported in Figure 3. Please add panel C in Figure 3.

7.    The discussion part is too long and unfocused. Please revise it.
